# The Importance of Immunohistochemical Heterogeneous Expression of MMR Protein in Patients with Colorectal Cancer in Stage II and III of the Disease

**DOI:** 10.3390/medicina59030489

**Published:** 2023-03-02

**Authors:** Tijana Denčić, Aleksandar Petrović, Maja Jovičić Milentijević, Goran Radenković, Marko Jović, Nikola Živković, Sonja Šalinger, Branko Branković, Aleksandra Veličkov, Ivan Ilić

**Affiliations:** 1Department of Pathology, Faculty of Medicine, Clinical Centre Niš, University of Niš, 18 000 Niš, Serbia; 2Department of Histology and Embryology, Faculty of Medicine, University of Niš, 18 000 Niš, Serbia; 3Department of Internal Medicine and Patient Care, Faculty of Medicine, Clinical Centre Niš, University of Niš, 18 000 Niš, Serbia; 4Department of Surgery, Faculty of Medicine, Clinical Centre Niš, University of Niš, 18 000 Niš, Serbia

**Keywords:** heterogeneous nuclear expression, mismatch repair protein, colorectal cancer, immunohistochemical analysis

## Abstract

*Background and objectives*: In patients with colorectal cancer (CRC), heterogeneous expression of Mismatch repair (MMR) proteins can manifest itself in several different forms and is not such a rare phenomenon. Therefore, it is very important to recognize the nuclear expression of MMR proteins of different MMR status in order to avoid false positive or false negative results. The aim of this study was to determine the frequency and distribution of heterogeneous expression of MMR proteins in patients with stages II and III of the disease as well as its association with clinical, demographic and pathological characteristics of CRC in relation to proficient and deficient expression of MMR proteins. *Material and Methods*: The study included 104 cases of colorectal cancer obtained from surgical colectomy material in stages II and III of the disease. *Results:* From a total of 104 patients with colorectal cancer, immunohistochemical analysis of the expression of all four MMR proteins showed that heterogeneous expression of MMR proteins (as well as deficient immunoreactivity of tumor cells) was present in 12 cases, while proficient expression of MMR proteins was detected in 80 tumors. *Conclusions*: Our study showed that the only independent predictors of the loss of MMR protein expression were younger patient age and right-sided anatomical location of the tumor. The study also established the existence of heterogeneous expression of MMR proteins in a non-negligible percentage of CRCs (11.5%), where heterogeneous nuclear expression of MMR proteins was described in several different forms.

## 1. Introduction

Globally, colorectal cancers (CRCs) are the third most common and the second most lethal malignancies, which today ranks them among the most important causes of morbidity and mortality worldwide [1]. CRCs belong to a heterogeneous group of tumors, and the modern classification of colorectal cancers classifies them into microsatellite unstable (MSI) and microsatellite stable (MSS) cancers [2,3,4].

Mismatch repair (MMR) proteins are proteins that participate in the repair of specific types of DNA errors that occur during cell replication. Given that there is a large number of MMR proteins, four MMR proteins (mutL homologue 1 (MLH1), mutS homologue 2 (MSH2), mutS homologue 6 (MSH6) and postmeiotic segregation increased 2 (PMS2)) have great clinical and therapeutic importance. CRCs with the MSI phenotype or with a deficient MMR (dMMR) status arise as a result of accumulation of mutations or due to epigenetic suppression, during which there is a disturbance in the normal MMR heterodimerization of proteins [5,6,7].

Assessment of dMMR status by means of immunohistochemical (IHC) expression analysis of these proteins or by means of polymerase chain reaction (PCR)-based analysis using capillary electrophoresis is of great diagnostic importance considering that colorectal cancers with dMMR status have different clinical, pathological, prognostic and therapeutic implications. About 20% of patients with dMMR colorectal cancer in stage II and III of the disease have a better clinical prognosis compared to patients with a proficient MMR status, while patients with dMMR colorectal cancer in stage II of the disease have a better clinical outcome but do not benefit from the application of adjuvant chemotherapy. A very small percentage of patients with metastatic dMMR CRC (5%) is associated with poor clinical outcome and resistance to standard chemotherapy. About 70% of patients with Lynch syndrome or hereditary nonpolyposis colorectal cancer are associated with dMMR status [8,9,10,11].

Deficient MMR status in sporadic CRCs, i.e., in CRCs with the MSI phenotype, represents an early event in the course of carcinogenesis, but it can also occur later in the course of tumor progression, resulting in tumor heterogeneity. In patients with CRC, heterogeneous expression of MMR proteins (hMMR) can manifest itself in several different forms and is not such a rare phenomenon. Distribution of heterogeneous MMR protein expression in patients with colorectal cancer was presented as intraglandular and/or zonal heterogeneity. Heterogeneous expression of MMR proteins as well as the protein pair was presented as areas with focal nuclear expression and/or areas with strong diffuse expression of proteins from the MMR group. Therefore, it is very important to recognize this phenomenon of nuclear expression of MMR proteins that can have different MMR statuses in order to avoid false positive or false negative results [12].

The aim of the study was to determine the frequency and distribution of heterogeneous expression of MMR proteins in the nuclei of colorectal cancer cells in patients in stage II and III of the disease and to determine its association with clinical, demographic and pathological characteristics of CRC in relation to proficient (pMMR) and deficient expression of MMR proteins.

## 2. Material and Methods

### 2.1. Sample Preparation and Collection

The study included 104 cases of colorectal cancer obtained from surgical colectomy material in stages II and III of the disease. Surgical resection of the colorectal region was performed at the Surgical Clinic at the University Clinical Center Niš. This study obtained the approval of the local Ethical Committee No.:12-3340-2/1. After fixation in a 10% formalin solution, representative tissue sections were molded into paraffin blocks and then stained with the classic hematoxylin–eosin method.

### 2.2. Sample Classification

From the archives of the Center for Pathology and Pathological Anatomy in Niš, by reviewing the register of pathohistological reports from the period 2018–2019, the following data were included in the analysis: tumor histological type, localization, sex, histological grade, lymphocyte response of the host, Crohn-like reaction, mucin secretion, peritumoral lymphovascular and perineural invasion and pathological stage of the disease. Based on the age of the patients, they were divided into two groups: under 50 years of age and over 50 years of age. Right-sided tumors involved the tumors localized in the cecum, ascending colon and transverse colon, whereas left-sided tumors developed in the descending colon, sigmoid colon and rectum. To determine the pathological stage of the disease defined by the tumor nodes metastases (TNM) WHO system, the depth of tumor invasion, status of lymph nodes and metastases were used.

To quantify tumor budding, from each microscope slide with cancer samples, the ×100 magnification field with the highest density of buds was identified, and it was then analyzed at ×200 magnification. The field with the largest number of buds was considered representative if it covered a field with the dimension of 0.785 mm^2^, which is defined by the International Tumor Budding Consensus Conference (ITBCC) system. In this study, a two-level system of the classification of tumor budding was used: low grade (0–9 tumor buds) and high grade (more than 10 tumor buds) [13].

### 2.3. Immunohistochemical Analysis

Immunohistochemical analysis was performed manually and carried out in the laboratories of the Department of Histology and Embryology of the Faculty of Medicine in Niš in accordance with the published methodology. Microtome sections were made from paraffin molds with a thickness of about 4 μm. After deparaffinization and hydration of the sections through xylene and a series of alcohols of decreasing concentrations of 100%, 95% and 80%, antigen unmasking was performed at 55 °C in a water bath for 30 min, followed by cooling at room temperature. After that, washing and blocking of endogenous peroxidase with 3% hydrogen was performed for ten minutes. Then the sections were washed in phosphate buffered saline (PBS) pH 7.4 three times for five minutes, after which the primary antibody was applied during incubation for 40 min at room temperature. After washing with PBS, the secondary antibody was applied with an incubation period of 20 min also at room temperature. Between each step, washing with PBS was done three times for five minutes. The presence or absence of nuclear expression of MMR proteins was examined using the immunohistochemical labeling technique: MLH1, clone ES05; PMS2, clone EP51; MSH2, clone FE11 and MSH6, clone EP49 (monoclonal antibodies were from the same manufacturer (Dako, Glostrup, Denmark), in ready-to-use form). The positive internal control on each section was represented by the surrounding unchanged mucosa, lymphocytes and stromal cells. Cases were defined as proficient MMR tumors (if positive tumor cell nuclear expression of all four MMR proteins was verified), dMMR (the loss of tumor cell immunoreactivity of at least one of the MMR proteins) or heterogeneous MMR tumors (areas with the loss of nuclear expression coexisting with areas of strong diffuse expression of proteins from the MMR group). En Vision visualization system with chromogen diaminobenzidine-DAB (Dako, Glostrup, Denmark) was used for visualization. Contrast staining, after immunohistochemical labeling, was performed with Mayer’s hematoxylin.

### 2.4. Statistical Analysis

The analysis of statistical data was performed using the software SPSS ver. 17.0 (SPSS Inc, Chicago, IL, USA). The studied groups were compared on the basis of demographic, clinical and histopathological characteristics of patients with colorectal cancer using the χ-square test and Fisher’s test. Categorical variables are given as frequencies (n) and in percentages (%). Univariate and multivariate logistic regression analysis examined the association between MMR-deficient cancers and factors of interest (calculated odds ratio (OR) value and its 95% confidence interval—95% CI).

## 3. Results

From a total of 104 patients with colorectal cancer, immunohistochemical analysis of the expression of all four MMR proteins showed that heterogeneous expression of MMR proteins (as well as deficient immunoreactivity of tumor cells (Figure 1) was present in 12 cases, while proficient expression of MMR proteins was detected in 80 tumors (Figure 2). Heterogeneous expression of PMS2 protein and heterodimeric pair MLH1/PMS2 was found in two tumors, MSH2/MSH6 was found in seven tumors (two of which also showed heterogeneity for MLH1/PMS2) and MSH6 was found only in one tumor. In seven out of twelve tumors micromorphologically, both forms of heterogeneous protein expression coexisted in the tumor tissue as intraglandular and zonal heterogeneity (Figure 3). Isolated intraglandular heterogeneous protein expression was found in two tumors (Figure 4), while zonal heterogeneity was present in three tumors (Figure 5).

Demographic (age and sex), clinical (tumor location) and pathological characteristics (TNM classification, histological tumor type, tumor grade, mucin secretion, peritumoral lymphocytic infiltration, Crohn-like reaction, tumor budding, metastases in regional lymph nodes and lymphovascular and perineural invasion) of colorectal cancer in relation to MMR protein expression (proficient, deficient and heterogeneous) are shown in Table 1.

Univariate regression analysis identified potential predictors of MMR protein expression (proficient < heterogeneous < deficient): age below 50 years, presence of mucin secretion, right-sided anatomical localization of the tumor, marked peritumoral lymphocytic infiltration and high-grade tumor budding (Table 2). Selected potential predictors of MMR protein expression were included in the multivariate model. Multivariate regression analysis yielded a significant model (F = 12.297, *p* < 0.001) that explains 39.7% of the variance of MMR protein expression. The only independent predictors of loss of expression were patients younger than 50 years (*p* < 0.001) and right-sided cancer localization (*p* > 0.01) (Table 2).

Using a univariate binary logistic regression, it was found that the presence of lymphovascular invasion increased the risk of heterogeneous protein expression by five times (*p* < 0.05) and was the only statistically significant predictor of heterogeneous MMR protein expression compared to proficient expression, with a share of 5.8–10.7% (Table 3).

The univariate binary logistic regression method was used to identify potential predictors of deficient MMR status in relation to heterogeneous tumor expression, which include age below 50 years, right-sided tumor localization, Crohn-like reaction and tumor budding. Multivariate analysis resulted in a statistically significant model (χ^2^ = 23.989, *p* < 0.001) that explained 63.2–84.3% of the variance in MMR protein expression (Table 4). However, none of the independent variables had an independent predictive value due to a small number of heterogeneous tumors obtained.

## 4. Discussion

Assessment of dMMR status is possible using PCR analysis as well as IHC analysis. Using immunohistochemical analysis of MMR proteins, deficient expression of MMR proteins implies a complete loss of nuclear expression of one or more MMR proteins, while intact nuclear expression of all four MMR proteins was classified as proficient. The loss of MSH2 or MLH1 protein function causes the loss of nuclear expression of the mentioned proteins, resulting in the loss of expression of their heterodimeric pairs (MSH6 and PMS2). However, the loss of expression of the individual MMR proteins MSH6 or PMS2 does not result in the loss of expression of their heterodimeric partners [14]. IHC analysis as well as PCR analysis represents a specific method (100%) of determining the MMR status. On the other hand, the sensitivity of this method in relation to PCR analysis is about 90% due to the existence of missense mutations in the MMR gene [15,16,17,18,19].

Although the interpretation of IHC results of MMR protein expression is relatively easy, a small number of papers report both false positive and false negative results. Incorrect immunohistochemical interpretation of MMR protein expression results is mainly due to technical artifacts. Poor fixation of tissues from surgical materials is one of the factors that affect IHC analysis. A small number of studies recommend IHC analysis on biopsy material precisely because of better tissue fixation, which results in more optimized preservation of antigens [20,21,22]. IHC analysis can also be affected by inadequate sample storage, antibody specificity, the presence of extensive areas of necrosis and the use of neoadjuvant chemotherapy and/or radiotherapy. It is recommended that if there is a loss of nuclear expression of the MMR protein in a CRC patient who received neoadjuvant therapy, IHC analysis should be repeated [23].

In order to ensure the highest possible reliability of the results, IHC analysis of MMR protein expression was repeated several times under different conditions for individual cases in our study. During the immunohistochemical procedure, antigen unmasking as well as the length of antibody exposure were modified. Based on the analysis of the entire sample, it was established that the tumor cells on the periphery of the preparation as well as on the invasive front showed strong and diffuse immunoreactivity, while the nuclear expression of cells that form neoplastic glandular formations in the central part of the tumor mass remained intact. Moreover, there was no observed necrosis.

In agreement with published results, our study found that the frequency of heterogeneous expression of MMR protein in patients with CRC was not so small and could have several different forms. Halvarsson et al. and Graham et al. described the distribution of heterogeneous MMR protein expression in patients with colorectal cancer and endometrial cancer. It was shown that in most cancers, there were areas with focal nuclear expression and/or areas with strong diffuse expression of proteins from the MMR group [24,25]. The study of Judge Joost et al. [26] provided criteria defining hMMR protein expression in tumor tissue, which was presented as intraglandular and/or zonal heterogeneity. Intraglandular heterogeneous immunoreactivity of tumor cells is defined as strong and focal nuclear expression of tumor cells within neoplastic glandular structures admixed with tumor cells showing the loss of MMR protein expression. Zonal heterogeneity of MMR proteins is presented as a diffuse loss of tumor cell immunoreactivity in multiple adjacent neoplastic glandular formations.

Initially, due to the positive nuclear expression of tumor cells, the study of hMMR protein expression was attached to the group of pMMR tumors; however, it was released as a separate group later. Our study showed 12 cases (%) of heterogeneous MMR protein expression in the entire sample. Heterogeneous expression of the PMS2 protein as well as the protein pair MLH1/PMS2 was observed in two tumors (16.7%), MSH2/MSH6 was observed in seven (58%) tumors (two of which also expressed heterogeneity for MLH1/PMS2) and heterogeneous nuclear expression of the MSH6 protein was observed only in one (8.3%) tumor. Intraglandular and zonal hMMR nuclear protein expression was registered in seven (58%) of a total of twelve cancers. Micromorphologically, intraglandular heterogeneous immunoreactivity of tumor cells was observed in two tumors (16.7%), while zonal heterogeneous expression was present in three tumors (25%). Our single-center study was also based on a comparison of demographic, clinical and pathological characteristics of CRC in relation to MMR protein expression (proficient, deficient and heterogeneous). Using the method of univariate regression analysis in our research, possible predictors of MMR protein expression (proficient, deficient and heterogeneous) were determined. These were included in the multivariate regression analysis, in which it was shown that the only independent predictors of the loss of expression were younger patients and right-sided anatomical localization of the tumor.

In the study by Natasha et al., heterogeneous MMR status was present in about 10% of CRC patients at a younger age. At the beginning of the study, these cases were included in the group of proficient tumors, but they were later separated because it was shown that at least three out of seven (42%) cases contained germline mutations [27]. A study by Tachon et al. showed heterogeneous expression of the heterodimeric pair of MLH1/PMS2 proteins in one tumor tissue that was later determined to be an MSI neoplasm [28]. Such observations were also confirmed in several previous studies [29,30,31]. In the study by Enrico Berrino et al., heterogenous expression of MMR proteins was identified in nine cases (4.5%) of hMMR colorectal cancer. The study demonstrated that in six out of nine colorectal cancers, at least one MMR protein/heterodimeric pair was not expressed, whereas the other MMR protein showed heterogenous expression at the same time [32].

The main limitation of the study is a small number of patients for whom manual IHC analysis was possible, which is not so rare in countries with limited resources. Based on the few published works, biopsy material provides a better and more optimal diagnostic quality. In our study, IHC analysis was performed on surgical material, which may be a limiting factor due to the presence of necrosis. Since information on MMR status is confirmed by the PCR method, a limitation of our study is the unavailability of this method. In accordance with technical possibilities, we believe that IHC analysis on biopsy material implies the analysis of a limited part of the tumor that may not include fields with alternative expression and fields of unchanged surrounding mucosa as a control. We believe that despite the limitations of the study, the surgical material we used, in our circumstances, can provide valuable data, although there is a need for further testing.

## 5. Conclusions

Our single-center study showed that the only independent predictors of the loss of MMR protein expression were younger patient age and right-sided anatomical location of the tumor. The study also established the existence of heterogeneous expression of MMR proteins in a non-negligible percentage of CRCs (11.5%), where heterogeneous nuclear expression of MMR proteins was described in several different forms. Lymphovascular invasion was the only independent predictor of heterogeneous versus proficient MMR protein expression. In order to further test MMR status, we suggest that such heterogeneous patterns of MMR protein expression by IHC analysis must be taken into consideration in order to ensure an accurate classification of MMR status.

## Figures and Tables

**Figure 1 medicina-59-00489-f001:**
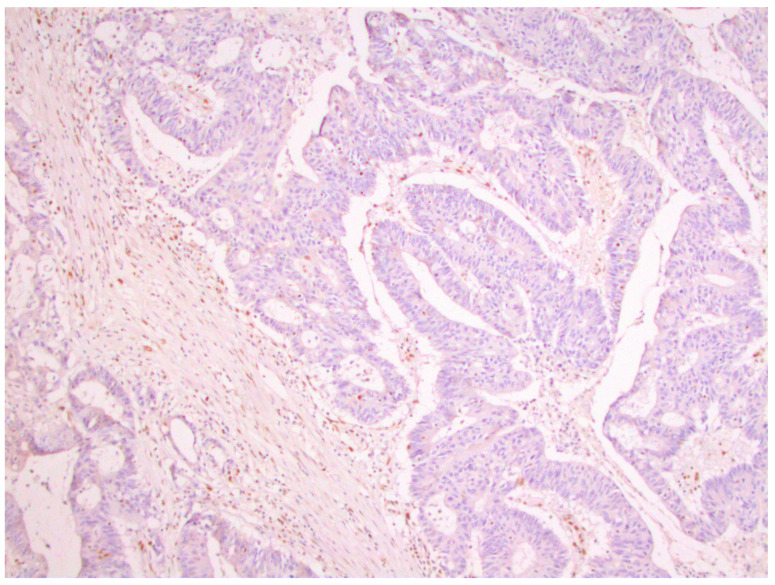
Deficient nuclear expression of MLH1 protein in colorectal cancer, ×100.

**Figure 2 medicina-59-00489-f002:**
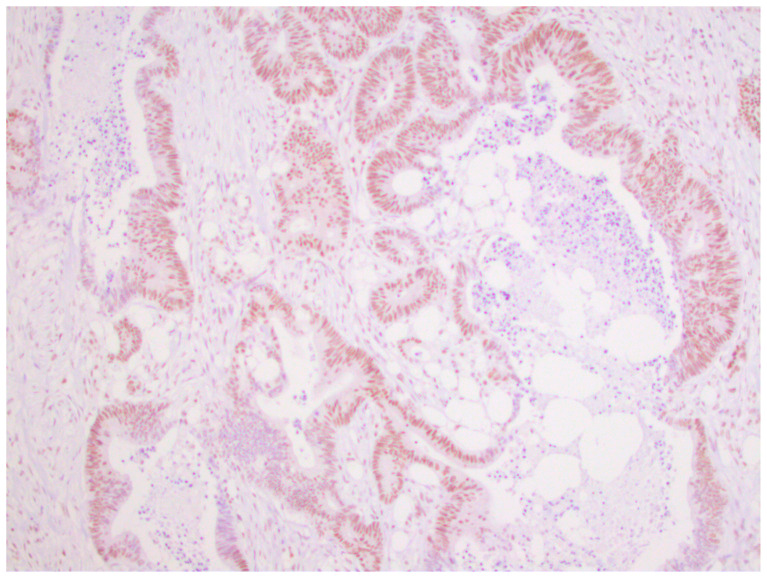
Proficient nuclear expression of MSH2 protein in colorectal cancer, ×100.

**Figure 3 medicina-59-00489-f003:**
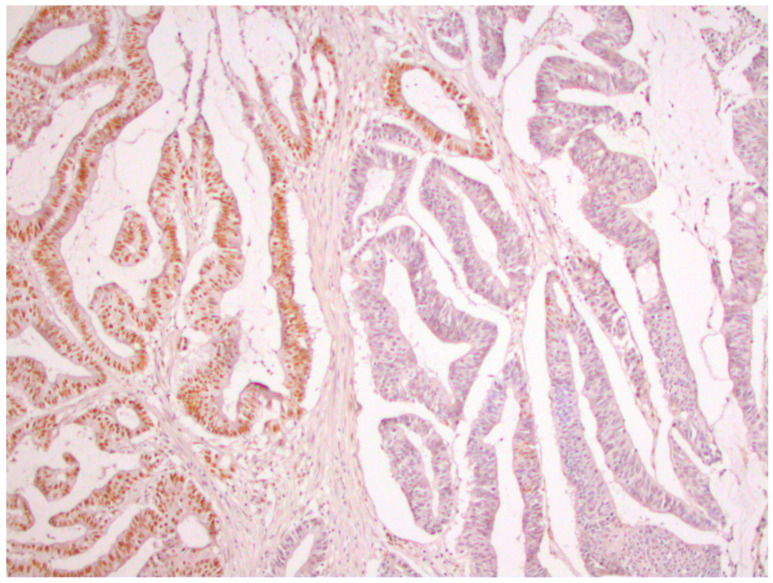
Zonal and intraglandular distribution of heterogeneous nuclear expression of MLH1 protein in colorectal cancer, ×100.

**Figure 4 medicina-59-00489-f004:**
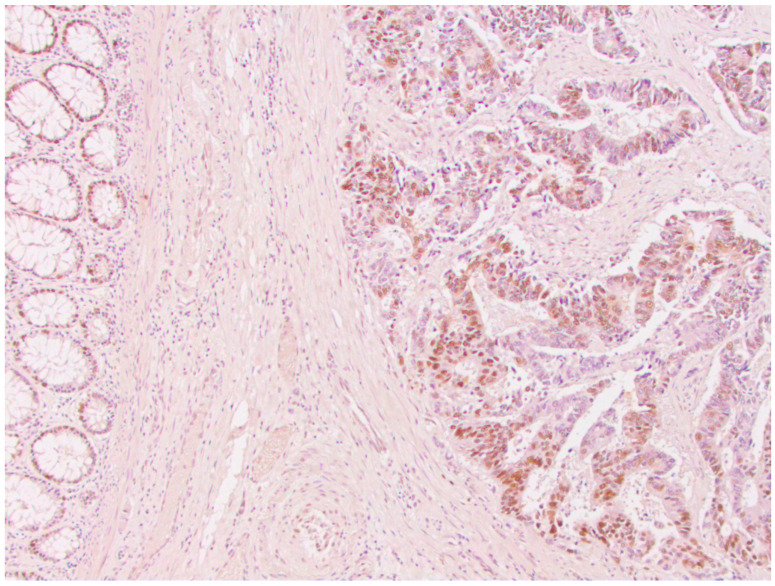
Intraglandular distribution of heterogeneous nuclear expression of MSH2 protein in colorectal cancer, ×100.

**Figure 5 medicina-59-00489-f005:**
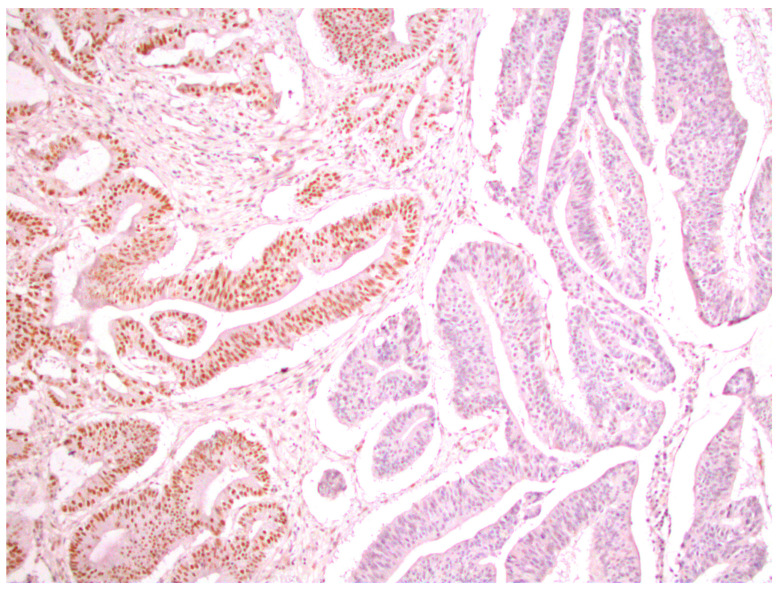
Zonal distribution of heterogeneous nuclear MLH1 protein expression in colorectal cancer, ×100.

**Table 1 medicina-59-00489-t001:** Demographic, clinical and pathological characteristics of patients with colorectal cancer in relation to MMR protein expression (proficient, deficient and heterogeneous).

Characteristics	Proficient MMR (n = 80)		Deficient MMR (n = 12)		Heterogenous MMR (n = 12)		Total (n = 104)		*p* Value
Age									
≤50	5	6.3%	9	75.0%	1	8.3%	15	14.4%	0.000
>50	75	93.8%	3	25.0%	11	91.7%	89	85.6%	
Sex									
Female	33	41.3%	7	58.3%	6	50.0%	46	44.2%	0.492
Male	47	58.8%	5	41.7%	6	50.0%	58	55.8%	
Tumor localization									
Right-sided	17	21.3%	10	83.3%	5	41.7%	32	30.8%	0.000
Left-sided	63	78.8%	2	16.7%	7	58.3%	72	69.2%	
TNM stage of the disease									
Stage III	40	50.0%	7	58.3%	8	66.7%	55	52.9%	0.516
Stage II	40	50.0%	5	41.7%	4	33.3%	49	47.1%	
Tumor type									
Conventional adenocarcinoma	60	75.0%	10	83.3%	6	50.0%	76	73.1%	0.133
Mucinous adenocarcinoma	20	25.0%	2	16.7%	6	50.0%	28	26.9%	
Tumor grade									
High-grade	43	53.8%	9	75.0%	10	83.3%	62	59.6%	0.077
Low-grade	37	46.3%	3	25.0%	2	16.7%	42	40.4%	
Mucin secretion									
Present	22	27.5%	7	58.3%	6	50.0%	35	33.7%	0.048
Absent	58	72.5%	5	41.7%	6	50.0%	69	66.3%	
Peritumoral lymphocyte infiltration									
Strong	31	38.8%	8	66.7%	8	66.7%	47	45.2%	0.055
Weak	49	61.3%	4	33.3%	4	33.3%	57	54.8%	
Crohn-like reaction									
Positive	11	13.8%	8	66.7%	2	16.7%	21	20.2%	0.000
Negative	69	86.3%	4	33.3%	10	83.3%	83	79.8%	
Tumor budding									
High-grade	16	20.0%	9	75.0%	1	8.3%	26	25.0%	0.000
Low-grade	64	80.0%	3	25.0%	11	91.7%	78	75.0%	
Lymphovascular invasion									
Present	39	48.8%	8	66.7%	10	83.3%	57	54.8%	0.055
Absent	41	51.3%	4	33.3%	2	16.7%	47	45.2%	
Perineural spread									
Present	37	46.3%	6	50.0%	6	50.0%	49	47.1%	0.949
Absent	43	53.8%	6	50.0%	6	50.0%	55	52.9%	
Metastases in regional lymph nodes									
Present	40	50.0%	7	58.3%	8	66.7%	55	52.9%	0.516
Absent	40	50.0%	5	41.7%	4	33.3%	49	47.1%	

Data were presented as frequencies (n) and in percentages (%); *p* < 0.001; tumor nodes metastases, TNM.

**Table 2 medicina-59-00489-t002:** Univariate and multivariate regression analysis.

**Univariate Regression Analysis**	***p* Value**	**OR**	**(95% CI of OR)**
Age ≤ 50 years	<0.001	1.076	0.762–1.389
Female sex	0.238	0.159	0.107–0.424
Right-sided tumor localization	<0.001	0.628	0.369–0.888
TNM stage of the disease (III)	0.394	0.114	0.151–0.379
Mucinous adenocarcinoma	0.927	0.015	0.284–0.314
High-grade tumor	0.054	0.261	−0.005–0.527
Presence of mucinous secretion	0.015	0.340	0.067–0.613
Expressed peritumoral lymphocyte infiltration	0.024	0.300	0.040–0.560
Crohn-like reaction (positive)	<0.001	0.640	0.334–0.946
High-grade tumor budding	0.001	0.513	0.223–0.802
Metastases in regional lymph nodes (present)	0.394	0.114	0.151–0.379
Lymphovascular invasion (present)	0.069	0.243	0.019–0.506
Perineural spread (present)	0.766	0.040	0.226–0.306
**Multivariate Regression Analysis**	***p* Value**	**OR**	**(95% CI of OR)**
Age ≤ 50 years	<0.001	0.875	0.552–1.198
Presence of mucinous secretion	0.706	0.046	−0.195–0.286
Right-sided tumor localization	0.003	0.393	0.137–0.648
Expressed peritumoral lymphocyte infiltration	0.939	−0.010	−0.264–0.245
Crohn-like reaction (positive)	0.228	0.201	−0.128–0.530
High-grade tumor budding	0.704	0.054	−0.229–0.337

Odds ratio, OR; confidence interval, CI.

**Table 3 medicina-59-00489-t003:** Univariate binary logistic regression analysis.

Univariate Regression Analysis	*p* Value	OR	(95% CI of OR)
Age ≤ 50 years	0.786	1.364	0.145–12.788
Female sex	0.569	1.424	0.422–4.805
Right-sided tumor localization	0.132	2.647	0.746–9.393
TNM stage of the disease (III)	0.288	2.000	0.557–7.177
Metastases in regional lymph nodes (present)	0.288	2.000	0.557–7.177
Mucinous adenocarcinoma	0.082	3.000	0.869–10.363
High-grade tumor	0.070	4.302	0.886–20.898
Presence of mucinous secretion	0.123	2.636	0.768–9.051
Expressed peritumoral lymphocyte infiltration	0.078	3.161	0.877–11.390
Crohn-like reaction (positive)	0.787	1.255	0.242–6.507
High-grade tumor budding	0.349	0.364	0.044–3.027
Lymphovascular invasion (present)	0.040	5.256	1.082–25.525
Perineural spread (present)	0.808	1.162	0.345–3.913

Odds ratio, OR; confidence interval, CI.

**Table 4 medicina-59-00489-t004:** Univariate and multivariate binary regression analysis.

**Univariate Regression Analysis**	***p* Value**	**OR**	**(95% CI of OR)**
Age ≤ 50 years	0.005	33.000	2.909–374.311
Female sex	0.682	1.400	0.279–7.016
Right-sided tumor localization	0.045	7.000	1.044–46.949
TNM stage of the disease (III)	0.674	0.700	0.133–3.684
Metastases in regional lymph nodes (present)	0.674	0.700	0.133–3.684
Mucinous adenocarcinoma	0.096	0.200	0.030–1.329
High-grade tumor	0.617	0.600	0.081–4.447
Presence of mucinous secretion	0.682	1.400	0.279–7.016
Expressed peritumoral lymphocyte infiltration	1.000	1.000	0.183–5.460
Crohn-like reaction (positive)	0.020	10.000	1.444–69.262
High-grade tumor budding	0.005	33.000	2.909–374.311
Lymphovascular invasion (present)	0.353	0.400	0.058–2.770
Perineural spread (present)	1.000	1.000	0.202–4.955
**Multivariate Regression Analysis**	***p* Value**	**OR**	**(95% CI of OR)**
Age ≤ 50 years	0.997	9.863 × 10^15^	0.000–∞
Right-sided tumor localization	0.997	6.014 × 10^15^	0.000–∞
Crohn-like reaction (positive)	0.998	120,003,272.9	0.000–∞
High-grade tumor budding	0.997	196,797,989.8	0.000–∞

Odds ratio, OR; confidence interval, CI.

## Data Availability

The data presented in this study are available on request from the corresponding author.

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
