# Peer review of "The Importance of Immunohistochemical Heterogeneous Expression of MMR Protein in Patients with Colorectal Cancer in Stage II and III of the Disease"

_medicina, 2023, doi:10.3390/medicina59030489_

Round 1

Reviewer 1 Report

The paper presents the results of immunohistochemical analysis of MMR protein expression in tumour tissues obtained from 104 patients with CRC. The paper is well written and the results are convincing. The remarks are as follows: - it is necessary to refresh the literature with more recent papers; - it is necessary to check systematic use of the full name of the phenomenon and abbreviations (for example in the Introduction chapter, paragraph 2, the abbreviation dMMR was used, and later in paragraph 3 the full name of the phenomenon and its abbreviation is given, this should be systematically changed in this and all other cases); - it is necessary to add images of immunohistochemical analysis of MMR protein expression in tissues that are MMR negative; - it would be useful to add data on the MSI/MSS status of the tumor; - paragraph 4 in the Introduction should be written more clearly.

Author Response

Dear Sir

Thank you for your comment. We are grateful for your consideration of this manuscript, and we also very much appreciate your suggestions, which have been very helpful in improving the manuscript.

Point 1. Literature data is correct.

Point 2. Abbreviation and the full name of the phenomenon is correct.

Point 3. Figure 1,2,3,4, and 5 are correct.

Point 5. In the introduction paragraph 4 is written more clearly.

Point 6. We performed IHC testing, because this method is not a routine procedure in our country, due to low fund (prospective part of our study). Because of low fund, we are not able to testing colorectal cancer of MSI/MSS status.

We hope that we have successfully revised the manuscript according to the valuable comments of the reviewers. The publication of our manuscript in your journal would be of immense importance for our future professional career.

Reviewer 2 Report

Materials and methods section

1. Author should separate the methods into 4 subsections: 2.1 Sample preparation and collection, 2.2 Sample classification, 2.3 Immunohistochemical analysis, and 2.4 Statistical analysis.

2. Why was not stage I of colorectal cancer used in this experiment?

Result section

1. What is a tumor location? Could you define this term? Did you mean the left/right side of the slide or the body in which the tissue was harvested?

2. In the table 1, the authors mentioned the location (right and left sided tumor), and it showed that total of MMR were highly found on the left side. Why was only right-sided tumor selected? 

3. Author look at table 2 be careful.

4. In the table 1, the total absence of perineural spread is not shown. This data should be added.

5. In the table 4, some data in this table is absent. Please check again.

Author Response

Dear Sir

We are grateful for your consideration of this manuscript, and we also very much appreciate your suggestions, which have been very helpful in improving the manuscript. We also thank the reviewers for their careful reading of our text.

All the comments we received on this study have been taken into account in improving the quality of the article, and we present our reply to each of them separately.  

Materials and metods section

Point 1. Materials and methods section is correct.

Point 2. We performed IHC testing, because this method is not a routine procedure in our country, due to low fund (prospective part of our study). We tried to implement guidelines suggested the diagnostic procedure for stage II, but also to look at the stage III tumor samples. We could not manage multicentric study, and we tried to emphasize the importance of this method not only in diagnostic but also in the therapeutic point of view. Adjuvant therapy was administered to almost of two third of patients, and it was useless and possibly harmful in a patients with stage II colorectal cancer. It was the reason why we not used patients with colorectal cancer in stage I of the disease.

Result section

Point 1. The right-sided tumors involved the tumors localized in the cecum, ascending colon, and transverse colon, whereas the left-sided tumors developed in the descending colon, sigmoid colon, and rectum.

Point 2. We collected 104 cases of colorectal cancer in the stage II and III of disease for which we had all available data from the period 2018-2019.

Point 3. Table two is correct.

Point 4. Table 1 is correct.

Point 5. Table 4 is correct.

We hope that we have successfully revised the manuscript according to the valuable comments of the reviewers.
